# PROTEIN DESIGNER BASED ON SEQUENCE PROFILE USING ULTRAFAST SHAPE RECOGNITION

## ABSTRACT

The process of designing proteins with specified structure and function, which can deepen our understanding of living systems and facilitate the fight against disease, involves a critical component known as sequence design. With the continuous development of deep learning, existing methods have shown excellent performance in protein sequence design. However, most of them focus on optimizing the network architecture to improve performance, while ignoring the explicit biochemical features of proteins. Observing the remarkable success achieved through structural templates and pre-trained knowledge in protein structure prediction, we explored whether similar sequence patterns and representations of underlying structural knowledge can be used in protein sequence design. In this work, we proposed SPDesign, a method for protein sequence design based on sequence profile using ultrafast shape recognition. For an input backbone structure, SPDesign utilizes ultrafast shape recognition vectors to search for similar protein structures (structural analogs) in the PAcluster80 structure library. It then extracts the sequence profile from the analogs through structural alignment. Along with structural pre-trained knowledge and geometric features, they are further condensed to provide reliable sequence patterns for an improved graph neural network. Experimental results show that SPDesign significantly outperforms the state-of-the-art methods on CATH 4.2 benchmark, such as LM-Design and Pifold, leading to 11.4% and 15.54% accuracy gains in sequence recovery rate, respectively. Encouraging results have been achieved on the TS50 and TS500 benchmarks, with performance reaching 68.64% and 71.63%, respectively. Particularly noteworthy is that our method also achieved significant performance on de novo designed proteins and orphan proteins that are close to practical application scenarios. Finally, the structural modeling verification experiment shows that the sequences designed by our method can fold into the native structures more accurately.

## 1 INTRODUCTION

Proteins with distinct structures exhibit diverse functions, and this functional diversity makes them play important roles in biological processes (Defresne et al., 2021). Protein design is the rational design of new active and functional protein molecules, which helps drug development, vaccine design, and reveals the basic principles of protein function (Marshall et al., 2019). As an important part of protein design, sequence design focuses on predicting the amino acid sequence that will fold into a specific protein structure (Pearce & Zhang, 2021; Ovchinnikov & Huang, 2021; Ding et al., 2022). With the continuous progress of deep learning, a series of different network architectures have been integrated into protein sequence design. These diverse frameworks have led to the development of innovative approaches, driving advancements in the field (Tan et al., 2023; Wang et al., 2018; Chen et al., 2019; Strokach & Kim, 2022; Karimi et al., 2020; Huang et al., 2023). Recently, the graph neural network stands out among these frameworks due to its excellent performance and superior compatibility with molecules in structure. As a result, many excellent methods have been proposed (Dauparas et al., 2022; Gao et al., 2023b; Hsu et al., 2022; Gao et al., 2022; Jing et al., 2021; Gao et al., 2023a). However, it is worth noting that most of these methods mainly focus on optimizing the network to improve performance, which might ignore the impact of protein features on accuracy.

Protein sequence design and structure prediction are inverse applications of each other, sharing certain similarities. Throughout the development of protein structure prediction, structural templates

have played an undeniably important role. Most structure prediction methods (Jumper et al., 2021; Baek et al., 2021; Lin et al., 2022; Chowdhury et al., 2022; Zhao et al., 2021) leverage templates as prior structural knowledge to enhance accuracy of protein structure prediction. Similarly, we can apply a similar sequential prior knowledge to protein sequence design. Following to the principle of template making in protein structure prediction, we search the structure library for proteins with similar structures to the input backbone (structural analogs), and extract the sequences corresponding to the matched parts of the analogs to create the sequence profile. Additional, pre-trained models are also widely used in the protein structure prediction (Rao et al., 2019; Brandes et al., 2022a) and protein sequence design (Alley et al., 2019; Hsu et al., 2022), yielding outstanding results.

In this work, we propose SPDesign, a method utilizing structural alignment to obtain the sequence profile to enhance the reliability of sequence design. In order to improve the search efficiency of structural analogs, SPDesign utilizes an improved ultrafast shape recognition algorithm Guo et al. (2022) that encodes both the input backbone structure and the center of clusters in the PAcluster80 structure library (Zhao et al., 2023) into USR-V vectors. This simplification transforms the complex shape comparison process into a similarity assessment between two vectors, significantly speeding up the search efficiency. The sequence profile contains sequence patterns from multiple structurally similar proteins, providing a reliable guidance. These patterns are then condensed through the statistical algorithm and a sequence pre-trained model respectively, and ultimately guide the design of the improved network together with other structural features (distance of backbone atoms, pre-trained knowledge of structure). Experimental results demonstrate that SPDesign achieves state-of-the-art sequence recovery rates on commonly used benchmarks (CATH 4.2 test set: 67.05%, TS50: 68.64%, TS500: 71.63%). In order to observe the performance of SPDesign in realistic application scenarios, we tested SPDesign on the de novo designed proteins obtained during practical design applications and the orphan proteins lacking sequence homologues provided by RGN2 (Chowdhury et al., 2022), and the results show that SPDesign achieves the best performance. Furthermore, compared with other methods, the sequences designed by SPDesign showed better ability to fold into native structures in the structure modeling verification experiment using high-precision, atomic-level prediction method ESMFold (Lin et al., 2022). In summary, our contributions include:

1. We propose a novel conception for extracting sequences from matched regions in structural analogs to create the sequence profile.

2. We propose SPDesign, a protein sequence design method that incorporates pre-trained knowledge and the sequence profile.

3. We propose an improved ultrafast shape recognition algorithm that encodes protein shapes into USR-V vectors, facilitating the efficient search for structural analogs.

4. Compared with other SOTA methods, experimental results show that the proposed method is the most superior.

## 2 RELATED WORK

**Methods in protein sequence design.** With the development of deep learning, many protein sequence design methods have been proposed. Based on their network architectures, these methods can be categorized into three groups: MLP-based, CNN-based, and GNN-based. SPIN and SPIN2 (Li et al., 2014; O'Connell et al., 2018) are early representative methods for sequence design using deep learning. They are based on the MLP network framework and integrate some protein structural features (torsion angle, backbone angle, neighborhood distance) to achieve a breakthrough in effect. However, due to the smaller network size, they are now generally have no advantage in effect. CNN-based models, like ProDCoNN and DenseCPD (Zhang et al., 2020; Qi & Zhang, 2020), can extract protein features in a higher dimension, thus achieving a higher sequence recovery rate. In contrast, these methods run relatively slowly due to the need for separate pre-processing and prediction for each residue (Gao et al., 2023b). The graph neural network (GNN) can effectively handle graph representations of molecular structures and has a fast running speed, so the GNN-based method can achieve a good balance between efficiency and accuracy. GraphTrans (Wu et al., 2021) introduces a graph attention and autoregressive decoding mechanism to improve the performance of the network; GVP (Jing et al., 2021) proposes a geometric vector perceptron to ensure the global equivariance of features; ProteinMPNN (Dauparas et al., 2022) calculates the ideal virtual atom $C_\beta$ as an additional

atom and employs the message-passing neural network (MPNN) to achieve a performance leap; Pi-fold (Gao et al., 2023b) enables the network to learn valuable atomic information independently, and proposes the PIGNN module, which achieves a high sequence recovery rate and greatly reduces the running time; LM-Design (Zheng et al., 2023) proves that language models with structural surgery are strong protein designers without using abundant training data.

**Structure library.** There are several widely used structural databases that can be used to search for structural analogs. PDB (Berman et al., 2002) is the most widely used protein database, containing 209,389 experimentally determined proteins, but without additional processing. CATH and SCOP (Sillitoe et al., 2020; Andreeva et al., 2019) classify proteins by structure on the basis of PDB, facilitating subsequent searches. In addition to the database containing experimental determined proteins, AlphaFold DB (Varadi et al., 2021) provides over 200 million virtual protein structures predicted by AlphaFold2. PAcluster80, provided by Pathreader (Zhao et al., 2023), is a composite database extracts protein composition from PDB and AlphaFold DB. PAcluster80 contains 56805 protein clusters, which were generated by clustering 106275 non-redundant PDB structures and 100912 AlphaFold DB structures with pLDDT$\geq$90 with 80% structural similarity.

**Pre-trained language models.** Pre-trained language models are used for protein structure prediction and protein sequence design. In protein structure prediction, UniRep and ProtBert (Alley et al., 2019; Brandes et al., 2022a) utilize Long Short-Term Memory (LSTM) and Transformer architecture respectively, to learn amino acid sequence representations from extensive sequence data. ESM2 (Lin et al., 2022) is the largest protein language model so far, which can achieve high-precision, atomic-level structure prediction using a single sequence. In protein sequence design, ProteinBERT (Brandes et al., 2022b) focus on capturing the local and global representation of proteins in a natural way, achieving excellent performance on multiple benchmarks covering a variety of protein properties. ESM-1F (Hsu et al., 2022) breaks through the limitation of the number of protein structures that can be determined experimentally, using the structure of 12 million sequences predicted by AlphaFold2 as an additional training data.

## 3 METHOD

In order to provide reliable sequence patterns, SPDesign uses a structural alignment algorithm to search for structural analogs and extracts the corresponding sequences to create the sequence profile. Next, the sequence patterns contained in the profile are processed by the statistical algorithm and a sequence pre-trained model, and consequently, it is converted into positional residue probability and sequence-pattern embedding. Along with other structural features (distance of backbone atoms, pre-trained knowledge of structure), the sequence-pattern embedding is then input into an improved graph neural network. Ultimately, guided by the positional residue probability, the designed sequence is obtained. The pipeline of SPDesign is shown in Figure 1.

### 3.1 STRUCTURE LIBRARY

We use the PAcluster80 (Zhao et al., 2023), which contains 56805 protein clusters and a total of 207,187 proteins, as the structure library used in the sequence profile production process. In order to ensure the fairness of the experiment, 40% of sequence redundancy is removed between all test sets and PAcluster80.

### 3.2 FEATURES

SPDesign employs a comprehensive set of four features, including sequence-pattern embedding and positional residue probability derived from the sequence profile obtained from structural analogs, as well as the distance of backbone atoms and pre-trained knowledge of structure. The acquisition process of all features is described in detail below.

**Sequence-pattern embedding and positional residue probability from sequence profile.** In order to obtain the proposed sequence profile, it is necessary to search for structural analogs of the input backbone in PAcluster80 structure library and then use high-precision structure alignment tool TM-align to perform fine alignment to obtain relevant sequences. Next, additional operations are applied to condense the features within the sequence profile, and finally extract the sequence-pattern

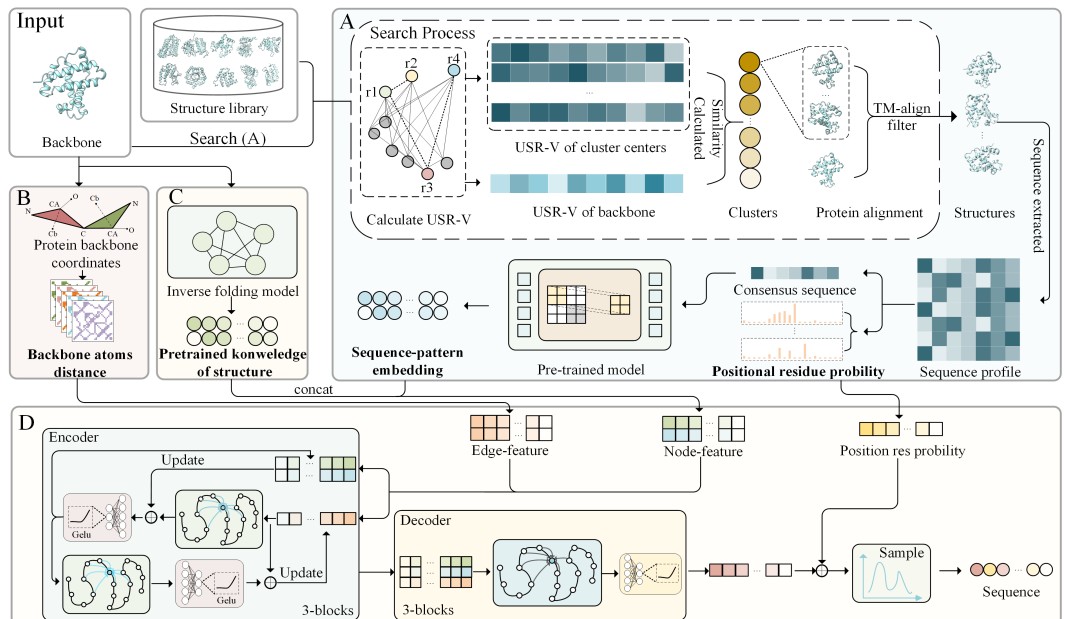

Figure 1: The workflow of SPDesign. (A) The input backbone structure uses USR-V to search for structural analogs in the structure library, and the sequences of the matched part are extracted to create the sequence profile. The consensus sequence, which is converted to sequence-pattern embedding by the pre-trained model, and positional residue probability are extracted from the profile. (B) Distance map of the backbone atoms. (C) Pre-trained knowledge of structure. (D) Features are aggregated and encoded by the network, and finally, decoded into a predicted sequence. All features in bold are used in the ablation experiment.

embedding and positional residue probability features. The detailed process is shown in Figure 1 (A).

However, it has been observed that during the search process for structural analogs, using TM-align is extremely time-consuming (one target requires more than 5 hours). To address this challenge, SPDesign divided the search process into two steps: first, using an improved ultrafast shape recognition algorithm to match the shapes between proteins to roughly and quickly screen the PAcluster80 structure library, and then use TM-align to perform fine structure comparison. As a result, this strategy improves search efficiency by 600 times.

In the first step, the USR-V of the input backbone structure is calculated and then its similarity with all cluster centers in PAcluster80 is compared to screen the most similar $k$ clusters. For the $i$-th residue ($r_i$) in the backbone, we find the nearest residue ($g_i$) and the farthest residue ($b_i$) of the current residue in Euclidean space. Additional, we find the residue ($f_i$) farthest from residue ($b_i$):

$$D_{r_i} = ||\vec{r_i} - \vec{r_j}||_2, j \in [1, L], j \neq i \tag{1}$$

$$\begin{cases} g_i = Ind(minD_{r_i}) \\ b_i = Ind(maxD_{r_i}) \end{cases} \tag{2}$$

$$D_{b_i} = ||\vec{b_i} - \vec{r_j}||_2, j \in [1, L], j \neq i \tag{3}$$

$$f_i = Ind(maxD_{b_i}) \tag{4}$$

where $\vec{r_i}, \vec{r_j}, \vec{b_j}$ represent the $C_\alpha$ coordinate vectors of residue $r_i, r_j, b_i$; $b_i, g_i, f_i$ correspond to the residues described above; $D_{r_i}, D_{b_i}$ respectively denotes the distance set from the residue $r_i, b_i$ to other residues; $L$ represents the total number of residues; $Ind$ signifies a function to obtain the subscript of the residue at the specified distance. After getting the four residues ($r_i, g_i, b_i,$ and $f_i$),

we proceed to calculate the average distance between these four residues and all residues within the protein.

$$d^{tp} = \frac{1}{L} \sum_{j=1}^{L} ||\vec{r_j} - \vec{tp}||_2, tp \in \{r_i, g_i, b_i, f_i\}, i \in [1, L] \tag{5}$$

For each individual residue, four distances can be obtained through the process described above. Generalizing the procedure to all residues in the protein, a distance matrix ($\mathbb{R}^{L \times 4}$) can be derived. Next, the distance matrix is transposed so that each row of it can be viewed as a distribution. Consequently, four different distances distributions $\{d^{r_i}\}_{i=1}^{L}, \{d^{g_i}\}_{i=1}^{L}, \{d^{b_i}\}_{i=1}^{L}, \{d^{f_i}\}_{i=1}^{L}$ can be obtained, corresponding to the four types of atoms $(r_i, g_i, b_i, f_i)$. Since a distribution is completely determined by its moments, for simplicity in subsequent calculations, we compute the first three moments of the distributions in order to characterize them as vectors (USR-V $\in \mathbb{R}^{1 \times 12}$) which encode the shape of the protein (Ballester & Richards, 2007). By calculating the mahalanobis distance between the USR-V of two proteins, the shape similarity of the two proteins can be obtained, illustrated as:

$$S(u_1, u_2) = \sqrt{(u_1 - u_2)^T \Sigma^{-1} (u_1 - u_2)} \tag{6}$$

where $u_1$ and $u_2$ represent the USR-V of the two proteins and $\Sigma$ is the covariance matrix between the vectors.

In the second step, to obtain more accurate matching information for sequence extraction, SPDesign utilizes TM-align to perform a comprehensive comparison between the input backbone and all structures within the chosen $k$ clusters. According to the TM-score obtained during the matching process, the aligned sequences of matched parts from the top $n$ proteins are extracted as the sequence profile.

On the basis of the sequence profile, the type and occurrence probability of amino acids appearing at each position are counted and used as the positional residue probability feature, which can provide some realistic basis for the sample process. At the same time, we take the most frequently occurring residues at each position to form a consensus sequence, and use the sequence pre-training model ESM2 to convert it into the sequence-pattern embedding.

**Distance of backbone atoms.** The atomic coordinates of the input protein backbone residues, which have a length of $L$, can be represented as $\{C_{\alpha_i}, C_i, N_i, O_i, C_{\beta_i} | 1 \le i \le L\}$, where $C_{\alpha_i}, C_i, N_i, O_i$ are backbone atoms and $C_{\beta_i}$ is an ideal atom computed from other atoms within the residue $i$ (Dauparas et al., 2022), as shown below:

$$C_{\beta_i} = \lambda_1 * \vec{a} + \lambda_2 * \vec{b} - \lambda_3 * \vec{c} + C_{\alpha_i} \tag{7}$$

$$\vec{b} = C_{\alpha_i} - N_i, \vec{c} = C_i - C_{\alpha_i}, \vec{a} = cross(b, c) \tag{8}$$

where $C_{\alpha_i}, C_{\beta_i}, N_i$ and $C_i$ represent coordinates of the corresponding atoms in residue $i$ and $\lambda_1, \lambda_2, \lambda_3$ represents three constant values. Next, the distances between these five atoms are calculated for all residues in protein and further transformed into a high-dimensional embedding using an advanced radial basis function (RBF).

**Pre-trained knowledge of structure.** The structural pre-trained model ESM-IF significantly expands the training set and exhibits superior versatility. Besides that, ESM-IF provides an accessible program interface that allows the translation of target protein backbone structure into a structural representation ($p \in \mathbb{R}^{L \times 512}$), termed pre-trained knowledge of structure, as shown in part (C).

### 3.3 NETWORK

As shown in Figure 1 (D), the architecture of SPDesign utilizes a message-passing neural network (MPNN), which can learn the characteristics of protein molecules directly from molecular graphs (Gilmer et al., 2017). In the architecture, the backbone is represented as a K-Nearest Neighbors (KNN) graph. The backbone graph $\mathcal{G}(\mathcal{V}, \mathcal{S}, \mathcal{E})$ consists of the node feature $\mathcal{V} \in \mathbb{R}^{L \times 128}$, edge feature $\mathcal{S} \in \mathbb{R}^{L \times N \times 128}$, and the set of edge between residues and their neighbors $\mathcal{E} \in \mathbb{R}^{L \times N}$, where N represents the number of neighbors.

**Encoding module.** The encoder consists of a stack of network layers with a hidden dimension of 128. For residue $i$, the node features of its neighboring nodes are fused with its edge features to

update the current node embedding and edge information in its propagation step (Dauparas et al., 2022), as shown below:

$$\mathcal{V}_i = g(f(p_i^{init})||g(t_i^{init}))$$

$$\mathcal{S}_i = f(s_i^{init})$$

$$\begin{cases} \mathcal{V}_m^{j\rightarrow i} = g(\mathcal{V}_i||(\mathcal{S}^{j\rightarrow i}||\sum_{j\in\mathcal{E}_{ij}}\mathcal{V}_j)) \\ \mathcal{V}_i = Norm(\mathcal{V}_i + Dropout(\mathcal{V}_m^{j\rightarrow i})) \end{cases} \tag{9}$$

$$d\mathcal{V}_i = g(\mathcal{V}_i)$$

$$\mathcal{V}_i = Norm(\mathcal{V}_i + Dropout(d\mathcal{V}_i))$$

where $f$ represents the operation of linear transformation; $g$ means the MLP layer consists of a linear layer alternating with the activation function; $Norm$ means the layer normalization; $Dropout$ indicates the dropout operation; $j \rightarrow i$ means the neighbors of residue i; $p^{init} \in \mathbb{R}^{L\times512}$ stands for pre-trained knowledge of structure; $t^{init} \in \mathbb{R}^{3L\times1280}$ represents sequence-pattern embedding; $s^{init}$ stands for the distance of backbone atoms and $||$ represents the concatenation operation.

Since general-purpose graph transformers cannot update edge features, critical information between residues may be ignored, affecting the performance of the network. To solve this problem, SPDesign uses an edge update mechanism (Dauparas et al., 2022), as shown below:

$$\begin{cases} \mathcal{V}_m^{j\rightarrow i} = g(\mathcal{V}_i||(\mathcal{S}^{j\rightarrow i}||\sum_{j\in\mathcal{E}_{ij}}\mathcal{V}_j)) \\ \mathcal{S}_i = Norm(\mathcal{S}_i + Dropout(\mathcal{V}_m^{j\rightarrow i})) \end{cases} \tag{10}$$

**Decoding module.** The decoder transforms the encoded data into output probabilities, following a principle similar to that of the encoder, same as shown in equation (9). In the decoding process, the node feature is integrated with the positional residue probability, ultimately yielding the residue probability feature corresponding to each position:

$$X = g(prp_i + f(\mathcal{V}_i)) \tag{11}$$

where $X$ is the resulting residue probability and $prp_i \in \mathbb{R}^{L\times20}$ is the positional residue probability.

## 4 EXPERIMENTS

### 4.1 DATASET

Using the same dataset as GraphTrans, GVP and Pifold (Wu et al., 2021; Jing et al., 2021; Gao et al., 2023b), 19746 proteins from CATH 4.2 (40% non-redundant) were divided into three parts, 17782 proteins for training, 844 for validation and 1120 for testing. In addition to the CATH 4.2 test set, we selected additional test sets to test the scalability of our method in practical application scenarios, such as the commonly used TS50 and TS500 datasets proposed by SPIN (Li et al., 2014), as well as the de novo designed proteins obtained during the practical design applications and the orphan proteins lacking sequence homologues provided by RGN2 (Chowdhury et al., 2022). For strict testing, we have removed 40% of the sequence redundancy between the sequence profile obtained from structural analogs and the input sequence.

### 4.2 PERFORMANCE

**Experiment results on CATH 4.2 test set.** The perplexity and sequence recovery rate results are presented in Table 1. Compared to ProteinMPNN and Pifold published recently, SPDesign shows a relatively significant improvement with a recovery rate that is 21.89% and 15.54% higher, respectively. Moreover, when compared to the latest method LM-Design, SPDesign's performance is still improved by 11.4%. Moreover, we also analyzed the amino acid type distribution used by SPDesign and the impact of different protein lengths on the CATH 4.2 test set, which are shown in Figure 2 (A), (B). The amino acid distribution used by our method is relatively consistent with the native amino acid distribution, and it has always been better than other methods in different length intervals.

Next, we analyzed the impact of consensus sequences derived from sequence profiles on the method. Assuming that the consensus sequence used by SPDesign for sequence-pattern embedding is treated as the designed sequence, we calculate the sequence recovery rate. Along with the performance obtained by SPDesign, the results are shown in Figure 2 (C). Clearly, the sequence recovery rate of the consensus sequence is very low, indicating a significant dissimilarity between the consensus sequence used by SPDesign and the native sequence. This phenomenon confirms that SPDesign improves performance not by relying on similar sequences but by incorporating additional information from external structural analogs and applying it correctly in the design process.

Table 1: Results on the CATH 4.2 test set. The best results are marked in bold and the best performing method is marked †.

| Model | Perplexity | | | Recovery | | |
|---|---|---|---|---|---|---|
| | short | single-chain | All | short | single-chain | All |
| SPDesign† | **4.71** | **4.65** | **2.43** | **48.21** | **47.06** | **67.05** |
| LM-Design | 6.77 | 6.46 | 4.52 | 37.88 | 42.47 | 55.65 |
| Pifold | 6.20 | 6.12 | 4.55 | 40.41 | 39.10 | 51.51 |
| ProteinMPNN | 7.04 | 7.86 | 5.25 | 29.20 | 27.33 | 45.16 |
| ESM-IF | / | / | / | 27.86 | 26.17 | 42.39 |
| GVP | 7.23 | 7.84 | 5.36 | 30.60 | 28.95 | 39.47 |
| GraphTrans | 8.39 | 8.83 | 6.63 | 28.14 | 28.46 | 35.82 |

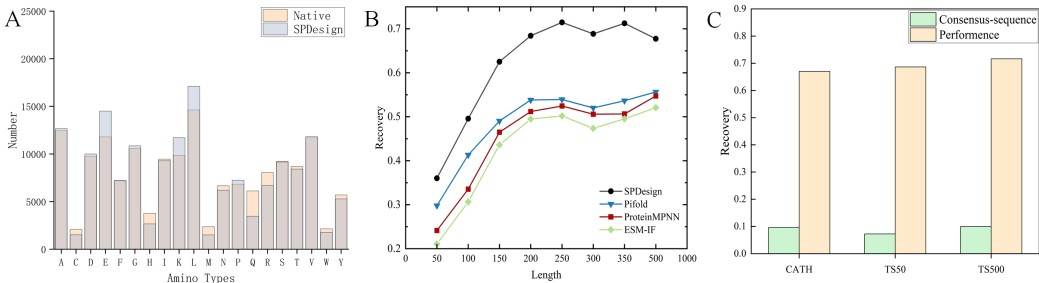

Figure 2: (A) Comparison of the amino acid type distribution selected by SPDesign with the native sequence amino acid type distribution. (B) Performance of methods at different protein lengths in the CATH 4.2 test set where the abscissa represents the interval from the previous scale to the current scale. (C) Comparison of the effects achieved by consensus sequence and SPDesign.

**Experiment results in orphan protein and de novo (designed) protein datasets.**

In order to test the performance fo SPDesign in real application scenarios, experiments and comparisons were conducted on orphan protein dataset and de novo (designed) protein dataset. The results shown in Figure 3 provide clear evidence that SPDesign outperforms other methods in situations where practical protein sequence design is required and when the target structure lacks sequence homologues. This finding aligns with previous inferences, indicating SPDesign's ability to extract valuable information from matched parts of structurally diverse, low-quality structures and effectively apply it to the sequence design process. In fact, this capability matches our original intention of creating models suitable for real sequence design applications, distinguishing it from other methods.

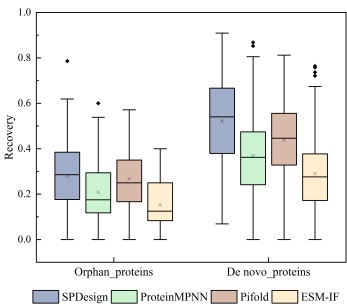

Figure 3: Results comparison on orphan proteins and de novo (designed) proteins.

**Experiment results on TS50 and TS500 datasets.** To evaluate the generality of SPDesign, in Table 2, we list the average perplexity and average recovery rates obtained on TS50 and TS500 datasets. Undoubtedly, SPDesign achieved better results than the other test sets, achieving the first breakthrough of 70% performance on TS500 with a recovery rate of 71.54%, and achieving a recovery rate of 68.64% on TS50. This might be the effect of protein length on model's performance. The protein lengths of the two test sets are mostly distributed between 100-300, which corresponds to the optimal performance range of the model. The excellent performance is consistent with previous experimental results, proving the powerful generalization ability of SPDesign.

Table 2: Results on the TS50 and TS500 datasets. The best results are marked in bold and the best performing method is marked †.

| Model | TS50 | | TS500 | |
|---|---|---|---|---|
| | Perplexity | Recovery | Perplexity | Recovery |
| SPDesign† | **2.72** | **68.64** | **2.54** | **71.63** |
| LM-Design | 3.50 | 57.89 | 3.19 | 67.78 |
| Pifold | 3.86 | 58.72 | 3.44 | 60.42 |
| ProteinMPNN | 3.93 | 54.43 | 3.53 | 58.08 |
| ESM-IF | / | 47.17 | / | 50.22 |
| GVP | 4.71 | 44.14 | 4.20 | 49.14 |
| GraphTrans | 5.40 | 43.89 | 4.98 | 45.69 |

## 4.3 ABLATION

To assess individual component impact on SPDesign, we conducted thorough feature ablation experiments, systematically removing each feature to observe their effects.

Table 3: Ablation study of features. Features marked with * are extracted from the sequence profile.

| | base | model[1] | model[2] | model[3] | model[4] | model[5] |
|---|---|---|---|---|---|---|
| Backbone atoms distance | ✓ | ✓ | ✓ | ✓ | ✓ | ✓ |
| Sequence-pattern embedding* | ✗ | ✓ | ✗ | ✓ | ✗ | ✓ |
| Pre-trained knowledge of structure | ✗ | ✗ | ✓ | ✓ | ✗ | ✓ |
| Positional residue probability* | ✗ | ✗ | ✗ | ✗ | ✓ | ✓ |
| Perplexity | 5.36 | 3.51 | 4.96 | 2.63 | 5.24 | 2.43 |
| Recovery | 44.60 | 58.76 | 48.81 | 66.85 | 45.12 | 67.05 |

The result presented in Table 3. The performance of model[1] indicates that sequence-pattern embedding, derived from the proposed sequence profile, is the most influential factor contributing to the model's performance, exhibiting a remarkable improvement of 14.16% over the base model. Undeniably, this result shows that sequence profile can serve as a valuable guidance to enhance SPDesign's performance significantly. Following closely, the pre-trained knowledge of structure and positional residue probability also exhibit positive impacts, show as model[2] and model[4], leading to enhancements of 4.21% and 0.52%, respectively. This result is expected because the pre-trained knowledge of structure provides a large amount of latent knowledge about the geometric structure, while positional residue probability can guide the model's sampling process to a certain extent.

After carefully observing the performance of model[1], model[2] and model[3], an interesting phenomenon is discovered. The combination of sequential pattern embedding and structural pre-trained knowledge improves the method performance by 22.25%, which is more than the sum of their individual improvements. This phenomenon might stem from the interaction effect between them,

such that their combination can provide more useful and richer information than using them independently.

## 4.4 STRUCTURAL MODELING VERIFICATION EXPERIMENT

To gain a more intuitive understanding of whether the designed sequences can accurately fold into the desired structures, we utilize the high-precision prediction method ESMFold to predict the structures of designed sequences. The predicted structure and the natural structure are evaluated by the structural similarity metrics TM-score and RMSD, which also represent the performance of methods. In order to make the result more realistic, the targets whose predicted structure generated by ESMFold using native sequence is very close to native structure on the CATH 4.2 test set are selected.

Table 4: Result of structural modeling verification experiment. The first row of the table indicates the threshold of TM-score, and the number of remaining proteins after screening is indicated in parentheses. The "native-sequence" signifies the reference performance using native sequence input. Due to conditional limitations, the table only lists the performance of several methods we have successfully reproduced.

| Model | TM-score >0.90(87) | | | TM-score >0.80(141) | | |
|---|---|---|---|---|---|---|
| | TM-score | RMSD | Recovery(%) | TM-score | RMSD | Recovery(%) |
| SPDesign | **0.940** | **1.216** | **69.42** | **0.893** | **1.545** | **75.77** |
| Pifold | 0.919 | 2.102 | 53.56 | 0.863 | 2.988 | 50.81 |
| ProteinMPNN | 0.937 | 1.631 | 50.61 | 0.884 | 2.475 | 45.11 |
| ESM-IF | 0.929 | 1.361 | 49.40 | 0.870 | 1.713 | 43.99 |
| native-sequence | 0.960 | 1.156 | 100 | 0.918 | 2.014 | 100 |

In Table 4, results at two different TM-score thresholds are presented. It can be seen that while SPDesign achieves a higher degree of sequence recovery, the structure predicted by the sequence is also more reasonable. Additionally, we observed that when the structural optimization space is limited, the improvement in sequence recovery rate leads to only minor optimizations in structure. However, on the other hand, the enhanced recovery rate might facilitate the folding of the sequence into the target structure more easily and stably within the biological context. Examples of target predicted using ESMfold are shown in supplementary material Figure 6.

## 5 CONCLUSION

In this work, we developed SPDesign, a method that incorporates sequence profile and pre-trained models for protein sequence design. To expedite search process for structural analogs, an improved ultrafast shape recognition algorithm is proposed for fast searching of structures with similar shapes. SPDesign outperforms other methods on well-known benchmarks (CATH 4.2 test set: 67.05%, TS50: 68.64%, TS500: 71.63%). Furthermore, we verified the encouraging performance of SPDesign on orphan and de novo proteins that are close to practical application scenarios.

The ablation experiment conducted in SPDesign confirms that sequence profile provides valuable guidance, significantly enhancing its performance. The application of SPDesign to antibody design will be our further research.

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

## A  APPENDIX

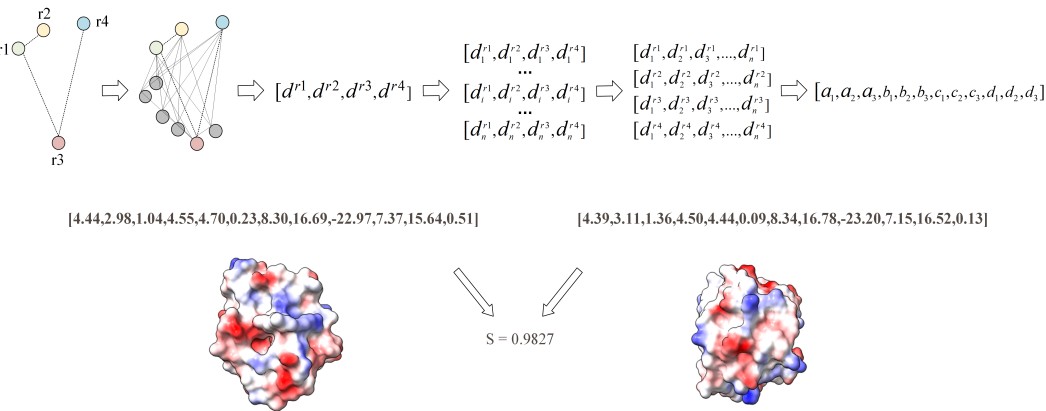

Figure 4: The main calculation process of USR-V.

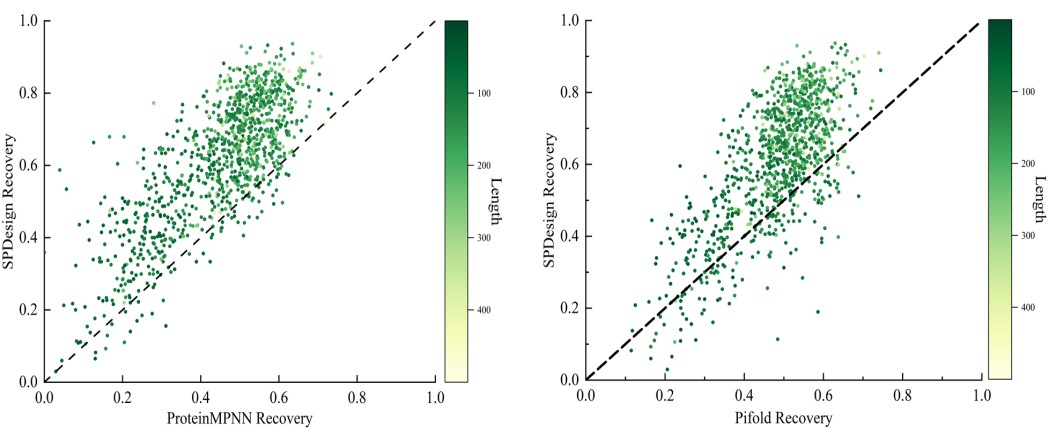

Figure 5: The performance comparison of SPDesign, ProteinMPNN, Pifold.

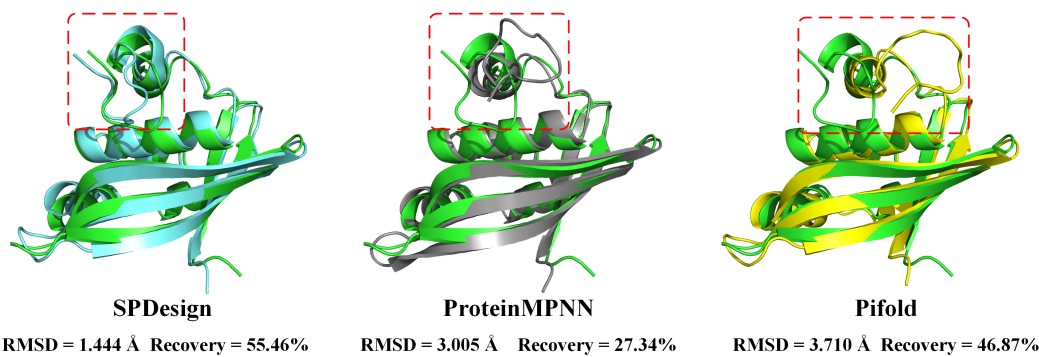

Figure 6: The Figure shows the results of SPDesign, ProteinMPNN and Pifold on the target (PDB 2jq5). It can be seen that the structure of SPDesign design is more reasonable than the others.

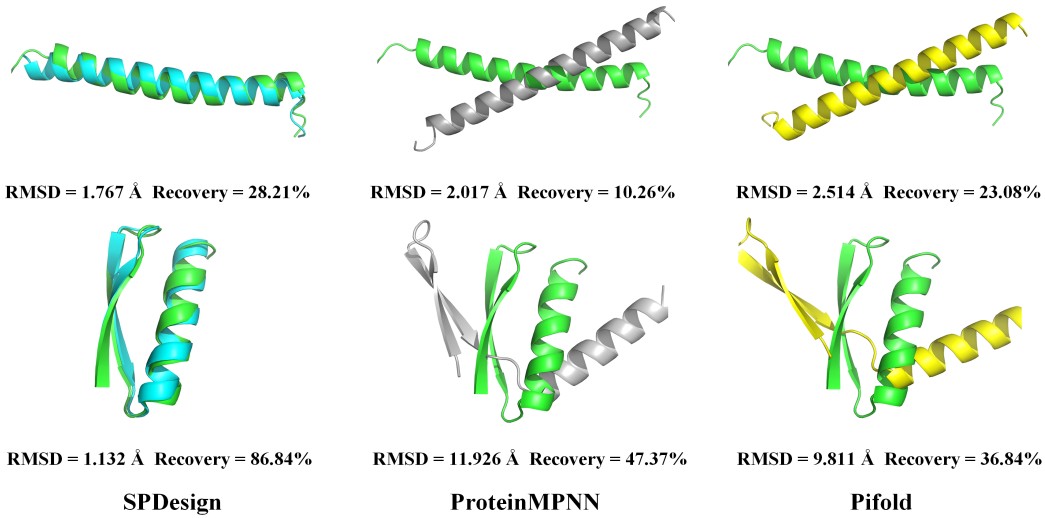

Figure 7: the results (PDB 7alo, 2nd3) of SPDesign, ProteinMPNN and Pifold on the orphan protein dataset and the de novo (designed) proteins dataset.

