# OpenReview forum: "PROTEIN DESIGNER BASED ON SEQUENCE PROFILE USING ULTRAFAST SHAPE RECOGNITION"
_ICLR.cc/2024/Conference — ICLR 2024 Conference Withdrawn Submission_

### Official Review · Reviewer_RyXy · 2023-10-23

**Soundness:** 3 good
**Presentation:** 2 fair
**Contribution:** 1 poor
**Rating:** 3
**Confidence:** 3

**Summary:**

This manuscript presents SPDesign, a workflow that integrates protein structure analogs into the input features to augment the performance of the message passing network in protein sequence design. The authors report enhanced empirical performance, which signifies a positive stride. However, the core contribution hinges on the application of a conventional idea to bolster feature engineering, diverging from novel machine learning methodologies. Upon revision, this manuscript may find a better fit in a bioinformatics-centric journal.

**Strengths:**

## Improved empirical performance
The authors have demonstrated improved empirical performance, which is commendable.

**Weaknesses:**

## Not open-source
The absence of open-source code hampers follow-up studies and comparisons, thereby limiting the community contribution.

## Limited technical advancement
It sounds like the primary distinction of SPDesign from earlier methods appears to lie in the feature input to the model, aligning the contribution more with traditional bioinformatics through feature engineering, rather than advancing machine learning techniques. Furthermore, as acknowledged by the authors, leveraging historical analogs for modeling guidance is a well-trodden idea in this field[1], which suggests a constrained conceptual contribution.

## Ill-defined task
It is noteworthy that the task of predicting protein sequences given a backbone structure is ambiguously defined. Conventionally, in protein design, the backbone and sequence are designed concurrently, underscoring a more common practice in the field[2].

## Lack of Accessibility to a General AI Audience
The manuscript employs certain acronyms such as USR-V and TM-align without elucidation, which may impede comprehension for a general ICLR audience.

Reference:
[1] Song, Y., DiMaio, F., Wang, R. Y. R., Kim, D., Miles, C., Brunette, T. J., ... & Baker, D. (2013). High-resolution comparative modeling with RosettaCM. Structure, 21(10), 1735-1742.
[2] Watson, J. L., Juergens, D., Bennett, N. R., Trippe, B. L., Yim, J., Eisenach, H. E., ... & Baker, D. (2023). De novo design of protein structure and function with RFdiffusion. Nature, 620(7976), 1089-1100.

**Questions:**

- Could the authors elucidate the acronym USR-V?
- An explanation of TM-align would be beneficial for clarity.

---

### Official Review · Reviewer_mFbR · 2023-10-30

**Soundness:** 2 fair
**Presentation:** 3 good
**Contribution:** 2 fair
**Rating:** 5
**Confidence:** 3

**Summary:**

This work presents a novel workflow and method for protein design (also termed inverse folding), an important and challenging problem in computational biology aiming to find an amino-acid sequence (or set of such sequences) that will fold into a predefined input 3D structure. Authors demonstrate that the combination of sequence profile information derived through search for structurally analogous proteins with their USR-V algorithm and SPDesign model utilising this information can lead to the substantial improvement in quality of the designed sequences assessed by native sequence recovery metric. Additionally authors verify performance of the workflow and quality of the designed sequences with harder external datasets and orthogonal structure prediction method.

**Strengths:**

Paper demonstrates that the combination of several existing ideas (addition of sequence profile information, pretrained models and GNNs for protein design) allows to improve over several state-of-the-art methods across investigated quality metrics. Additionally authors provide an ablation study allowing for deeper understanding of how each of the individual components affects final performance and verify their predictions in silico with orthogonal folding method. USR-V similarity search method described by the authors is a moderately small extension of the original papers (Ballester et al, 2007, J. Comput. Chem and Guo et al. 2022, Bioinformatics). Overall the paper is well written, structured and easy to follow. It touches on one of the fundamental problems in computational biology and research in this direction may have potential to enable accurate and rapid design of functional protein molecules for use in medicine or green industries.

**Weaknesses:**

- As one of the contributions of the paper authors mention an "improved ultrafast shape recognition algorithm (...), facilitating the efficient search for structural analogs". I believe that this statement is not properly supported by the data, in particular:

a) It's not clear how the method speed compares to state-of-the-art tools like FoldSeek (van Kempen et al, 2023, Nat Communications), 3D Zernike approach used in the PDB structural searches (Guzenko et al, 2020, Plos Comput Biol).

b) It's also not clear how efficient the method is compared to e.g. FoldSeek in extracting analogous structures. The pre-alignment step relies on global shape comparison which can prohibit from detection of some of sub-domain sized analogous fragments or fail to work with multidomain proteins (I also miss discussion of this in limitations of the authors approach).

I believe that extra experiments and comparisons are required to support the statements for this part or the text should be rewritten.

- SPDesign workflow is compared to the methods that do not take advantage of sequence profiles.

I agree with authors that combination of sequence profiles and GNNs for protein design can be considered as novel and interesting but there are already several available design methods that rely on the MSAs or PSSMs - e.g. Sgarbossa et. al, 2022, Elife or traditional Rosetta(Scripts) FavorSequenceProfile mover, to name a few. I believe that statements in the paper could be more convincing if authors add more MSA-related baselines. Also, to large extent, authors fail to discuss these more traditional but still large and meaningful body of work in the related work section.

- Lack of diverse metrics & analyses.

Protein design is a complex problem and frequently multiple solutions are equally good / stable for a given input backbone structure. Native sequence recovery is an important metric but doesn't tell the full story about the method properties and performance. I think it'd be valuable to understand e.g. how diverse are the sequences generated by the SPDesign for a given backbone, how closely designs resemble the natural sequence profiles, is the performance equal or better / worse for particular secondary structure elements or surface / buried residues? (see e.g. Ó Conchúir et al, 2015, Plos One, Ollikainen et al, 2013 Plos Comput Biol, Castorina et al, 2023, Bioinformatics for more examples of metrics / analyses)

**Questions:**

- In paragraph 4.1 the sentence "For strict testing, we have removed 40% of the sequence redundancy between the sequence profile obtained from structural analogs and the input sequence" is unclear to me - I suspect it means that the redundancy was removed at 40% sequence identity / similarity level but I'd like authors to confirm and perhaps rephrase this statement.

---

### Official Review · Reviewer_igsg · 2023-10-31

**Soundness:** 3 good
**Presentation:** 3 good
**Contribution:** 2 fair
**Rating:** 5
**Confidence:** 4

**Summary:**

SPDesign is a paper that leverages sequence profiles and pre-trained models for protein sequence design. By incorporating protein-specific biochemical features, it integrates sequence profile information from structurally similar analogs, protein geometry features, and pretrained knowledge of structures. The approach outperforms other SOTA methods on CATH4.2, TS50, TS500 benchmarks, demonstrating improved accuracy.

**Strengths:**

In addition to improved accuracy, the paper highlights two strengths.

1. It utilizes ultrafast shape recognition vectors (USR-V) for efficient searching of similar protein structures, significantly reducing computational time compared to TM-align.
2. The proposed method is applicable to orphan proteins and de novo protein design, making it suitable for practical design applications.
3. The paper is well-written and easy to follow.

**Weaknesses:**

The paper incorporates sequence profiles from similar analogs, but the network structure lacks novelty compared to recentworks. Additionally, the comparison of orphan protein and de novo protein experiments using box-whisker plots lacks specification on the datasets being compared.

**Questions:**

1. The network structure is questionable: part A is USR parts (referring to relevant papers); part B, D is ProteinMPNN structures, with almost no changes; part C is ESM and ESM-IF models. Is the structure only an integration of all these other models?

2. The paper heavily relies on mathematical deductions from other works, primarily USR and ProteinMPNN, with minimal modifications. This raises concerns about the originality and novelty of the research.

3.Have the authors tried any structural search methods other than USR?

4. Why should it be transformed into a consensus sequence instead of the entire sequence profile being input?

5. The authors fail to provide a clear explanation regarding the source and composition of the dataset in Figure 3, which can be quite confusing for readers.

6. Typo, e.g. 'probility' in Figure1.

---

### Official Review · Reviewer_VpDG · 2023-11-04

**Soundness:** 3 good
**Presentation:** 2 fair
**Contribution:** 2 fair
**Rating:** 5
**Confidence:** 4

**Summary:**

This paper proposes a protein sequence design method by incorporating sequence profiles from structural analogs along with other features into a graph neural network model. The idea of using ultrafast shape recognition to efficiently search a structure library for analogs to create a sequence profile is creative and offers a new way to provide sequence guidance. The proposed model SPDesign achieves strong results surpassing prior state-of-the-art methods on several benchmarks. The strengths are the novel incorporation of shape recognition and nice results on some benchmarks. However, the limitations outweigh the contributions of this work in its current form. The analog search process seems impractical, and the lack of thorough analysis and comparisons means the performance gains cannot be reliably attributed to the proposed approach.

**Strengths:**

(+) Novel concept of creating sequence profiles from analogs to provide guidance

(+) Significant performance gains demonstrated on benchmark datasets

(+) Ablation studies confirm value of sequence profile.

**Weaknesses:**

(-) More analysis is needed on the quality and diversity of analog structures found, and how this impacts performance. Are certain analog types more useful than others?

(-)  The search process for analogs seems computationally expensive. Can this be made more efficient? How sensitive is performance to the analog search parameters like number of clusters, analogs etc?

(-) Writing of the manuscript could be largely improved. The presentation could be tightened up and made more concise in places to improve clarity and flow.

**Questions:**

See weaknesses.